# Nutraceutical and Technological Properties of Buffalo and Sheep Cheese Produced by the Addition of Kiwi Juice as a Coagulant

**DOI:** 10.3390/foods9050637

**Published:** 2020-05-15

**Authors:** Andrea Serra, Giuseppe Conte, Leonor Corrales-Retana, Laura Casarosa, Francesca Ciucci, Marcello Mele

**Affiliations:** 1Department of Agriculture, Food and Environment, University of Pisa, via del Borghetto 80, 56124 Pisa, Italy; giuseppe.conte@unipi.it (G.C.); leocorretana@gmail.com (L.C.-R.); laura.casarosa@unipi.it (L.C.); francesca.ciucci88@gmail.com (F.C.); marcello.mele@unipi.it (M.M.); 2Center of Agricultural and Environmental Studies “E. Avanzi”, University of Pisa, via Vecchia di Marina, San Piero a Grado, 6-56122 Pisa, Italy; 3Research Center of Nutraceuticals and Food for Health, University of Pisa, via del Borghetto 80, 56124 Pisa, Italy

**Keywords:** milk clotting, cheese, kiwifruit, actinidin, nutraceutical properties

## Abstract

Kiwifruit is an interesting alternative to chymosin for milk coagulation. Although the clotting properties of actinidin (the proteolytic agent present in kiwi) have been widely investigated, little is known about the nutraceutical and organoleptic effects of kiwifruit on the characteristics of cheese. We investigated kiwifruit pulp, compared to calf rennet, in cheesemaking using sheep and buffalo milk. Although the kiwifruit extract showed a longer coagulation and syneresis time than calf rennet, it could nevertheless be exploited as a plant coagulant due to its positive effect on the nutraceutical properties. In fact, the sheep and buffalo cheese were higher in polyphenols and phytosterols than the cheese obtained using calf rennet. In addition, the nutraceutical properties were enhanced, with just a slight effect on the aroma of the cheese.

## 1. Introduction

Enzymatic milk coagulation is a key step in cheese manufacturing and involves the addition of chymosin (rennet), an aspartate proteinase that is active in the stomach of non-weaned calves [1], and which hydrolyses the link between amino acids 105 (methionine) and 106 (phenylalanine) of the k-casein. Given various social (i.e., veganism) and religious (Islam, Judaism) issues, which entail limiting or reducing the use of chymosin, new sources of coagulants are needed.

Proteolytic enzymes extracted from plants may be an interesting alternative to animal rennet in dairy technology. In fact, milk-clotting enzymes have been identified in various plant species, such as *Lactuca sativa* [2], *Albizia lebbeck*, *Helianthus annuus* [3] and *Cynara cardunculus* [4].

Actinidin (EC 3.4.22.14) is a cysteine protease from kiwifruit (*Actinidia deliciosa*) with a wide pH activity range (4–10) [5]. Lo Piero et al. [5] demonstrated that actinidin forms milk clots with the typical conditions used in cheese manufacturing (optimum activity at 40–42 °C, mildly acidic pH values). The preferred substrate for actinidin is β-casein, followed by k-casein, and the result of this hydrolysis is the production of a small number of larger peptides [5]. Saha and Hayashi [6] revealed that dairy products that use kiwifruit juice actinidin have a lower perceived off-flavour.

Exploiting kiwifruit in milk cheesemaking could meet the goals of circular agriculture through the use of undersized and/or damaged kiwifruits in a simple and economically sustainable procedure for the production of a clotting mixture. In addition, it could improve the nutritional/nutraceutical characteristics of the cheese, given that kiwifruit contains hydro-soluble components, such as vitamin C and polyphenols [7,8,9].

Several works have dealt with the clotting properties of kiwifruit. Katsaros et al. [10] reported that 1 mL juice rich in actinidin obtained from peeled, pulped and centrifuged kiwifruit corresponded to 0.42 ± 0.02 units, while 1 g of freeze-dried powder corresponded to 520 ± 20 units. Mazorra-Monzano et al. [11] compared kiwi, ginger and melon with chymosin and found that the kiwi extract produced the most similar curd in terms of texture to curd from chymosin. Lo Piero et al. [5] characterized the role of purified actinidin in casein proteolysis and reported that the activity for total casein was 129.6 U/mg and that the specific milk clotting activity (AC) is higher than the general one (AP); the AC/AP ratio (1.1) was similar to calf rennet (1.0) and much higher than microbiological rennet (0.22). Puglisi et al. [12] reported that kiwi juice shows twice the specific activity for total casein than purified actinidin of 299 U/mg, and speculated that, in kiwi, other proteases assist actinidin in the proteolysis process.

To the best of our knowledge, no data are available on the relationship between using kiwifruit extract in cheesemaking and the nutraceutical substances and flavour of cheese obtained with sheep and buffalo milk. We assessed the technological, nutraceutical and organoleptic properties of sheep and buffalo cheese made from kiwifruit extract compared with animal rennet.

## 2. Materials and Methods

### 2.1. Experimental Design

The study was carried out on 12 cheeses obtained from fresh and pasteurized milk, according to the following procedure: three buffalo milk cheeses were made with calf rennet (BM-C); three buffalo milk cheeses were made with kiwifruit extract (BM-K); three sheep milk cheeses were made with calf rennet (SM-C); and three sheep milk cheeses were made with kiwifruit extract (SM-K). The experiment was replicated twice in two independent batches. Milk was obtained from two neighbouring farms located in the province of Grosseto (southern Tuscany, Italy).

The kiwifruit extract was obtained by a modification of the method described by Katsaros et al. [10]. Briefly; 1 kg of kiwi pulp (*A. deliciosa*, cv. Hayward) were filtered through a cotton gauze. The filtrate was centrifuged (10,000 rpm for 15 min at 4 °C), and the supernatant was filtered using a 0.45 μm filter. The purified solution was lyophilized to obtain a powder rich in actinidin; the protein content of the powder was quantified by Bradford’s method [13].

The calf rennet used for the coagulation of the BM-C and SM-C cheeses was NATUREN ^®^ PLUS 215 (activity = 215 IMCU/mL; chymosin 63%; pepsin 37%) (CHR HANSEN, Hoersholm, Denmark).

Milk (200 mL) was clotted after the adjustment of the pH value by the addition of *Streptococcus thermophilus* CRV00 LYO 100 L starter (Santamaria srl, Burago di Molgara, Italy) (0.216 g per 100 mL) at 40 °C. Coagulants were added when the milk pH was 6.35–6.38. Calf rennet was diluted in distilled water to have a strength of 40 IMCU/mL. In order to have the same enzymatic activity (measured as Katsaros et al. [10]), for 200 mL of milk we used 80 mg of freeze-dried powder rich in actinidin.

After the milk had clotted, the curd was manually cut with a knife and stirred gently. Then, after about 10 min the cheeses were drained and put into moulds, which were turned and pressed manually. Finally, in order to eliminate the excess water, the miniature cheeses [11] were incubated at 40 ° C for 30 min. The cheeses were not salted. The weight of each cheese was recorded at the end of the cheesemaking and after 24 h. The syneresis of the cheeses was used as one of the indices of the level of coagulation, which was expressed as the volume of the whey (mL) per total mass (100 g) of the coagulated milk product. Cheese yields (initial and after 24 h) were calculated as curd weight (initial or after 24 h)/milk weight × 100.

### 2.2. Analysis

#### 2.2.1. Milk Clotting

Milk-clotting was evaluated visually when clotting began, in accordance with Uchikoba et al. [14]. During the visual evaluation, the time was measured between the addition of the coagulant solution and the first appearance of solid material against the background (CT, clotting time).

Milk-clotting properties were evaluated by a Formagraph ^®^ (Foss Electric, Hillerød, Denmark). Following the Formagraph ^®^ instructions, milk samples (10 mL) were heated to 35 °C, and 200 μL of a solution of rennet (NATUREN ^®^ PLUS 215-215 IMCU/mL; chymosin 63%; pepsin 37%) (CHR HANSEN, Hoersholm, Denmark) with a strength of 40 IMCU/mL was added; 10 replicates per run and for each sample milk were performed. The same conditions were applied to evaluate the kiwifruit extract; however, 200 μL of a 40mg/mL solution of freeze-dried powder in distilled water was used. Measurements were stopped thirty minutes after the addition of the enzyme.

The principle of lacto-dynamography is based on the control of the oscillation that is driven by an electromagnetic field created by a swinging pendulum. During milk clotting, a pendulum is immersed into the milk container. The greater the extent of the coagulation, the smaller the pendulum swing. This analysis provided measurements of the clotting time (r) in min, curd firming time (k20) in min, and curd firmness (A30) in mm [15].

#### 2.2.2. Physical and Chemical Analysis

Cheese samples were analysed in terms of moisture, protein, fat, and ash following official AOAC methods (AOAC, 2000). For the colour measurements, samples were placed on a standard white tile. Colour readings were taken at four randomly selected locations on the cranial surface of each piece to obtain a representative mean value. The cheese colour was measured in the CIE L*a*b* space (CIE, 19876) with an area diameter of 8 mm, including the specular component, and 0% UV, D65 standard illuminant, observer angle 10°, and a zero and white calibration using a Minolta CM 2006d spectrophotometer (Konica Minolta Holdings, Inc., Osaka, Japan). Lightness (L*), greenness (a*) and yellowness (b*) were recorded [16,17]. The colour parameters were used to calculate the total colour differences between cheeses obtained with calf rennet and cheeses obtained with kiwifruit, using the following formula: ∆E* = ((L*)^2^ + (a*)^2^ + (b*)^2^)^1/2^. Values were expressed as the mean ± standard deviation. Following Sanz [18], the colour differences for the human eye are not obvious if ∆E* < 1; not appreciable if 1 < E* < 3; and obvious if E* > 3.

Calcium, iron, sodium, magnesium and potassium were determined by flame atomic-absorption spectroscopy on an iCE 3000 series AA spectrophotometer (Thermo-Scientific, Waltham, MA, USA) equipped with a deuterium lamp as a background-correction system. An acetylene-air flame was used, while the gas flow rates, and the burner height were adjusted in order to obtain the maximum absorbance signal for each element. The organic matter of the samples (0.5 g) was put in a muffle-furnace at 450 °C for 24 h to obtain ash. When cool, the residue was dissolved in 1 mL nitric acid and the volume was diluted to 10 mL with water. The wavelength of the spectrometer was set at 422.7 nm for Ca, 589.6 nm for Na, 248.3 nm for Fe, 766.5 nm for K and 285.2 nm for Mg. The slit width was 0.7 nm for all elements, except Fe with 0.2 nm. The volumes and corresponding concentrations of the samples were selected within the linear range of the instrument used (at least five concentrations).

#### 2.2.3. Lipid Composition of Kiwifruit and Cheeses

Total lipids (TL) of kiwifruit and cheeses were extracted with a chloroform/methanol solution (2:1, v/v), following Rodriguez-Estrada et al. [19].

Unsaponifiable matter was obtained following Sanders et al. [20]. Briefly, 300 mg of TL were cold-saponified by adding 4.5 mL of ethanolic KOH (4.8% w/v) solution and incubated at room temperature for 12 h. The unsaponifiable matter was isolated by two washes with 4.5 mL of water and 9 mL of hexane. The non-polar phase (upper phase) was transferred into a fresh tube and dried by nitrogen gas. Finally, the samples were dissolved once again with 1 mL of methanol. Before saponification, 100 μL of a solution of dihydrocholesterol in chloroform (2 mg/mL) as internal standard for sterols were added to TL.

Sterols were then silylated adding a hexamethyldisilazane/chlorotrimethysilane/pyridine 2/1/5 v/v/v mixture, dried under a nitrogen stream and dissolved in 300 μL of n-hexane. The sterols were identified and quantified using a GC–FID (GC 2000 plus, Shimadzu, Columbia, MD, USA) equipped with a VF 1-ms apolar capillary column (25 m × 0.25 mm i.d., 0.25 μm film thickness; Varian, Palo Alto, CA, USA). A total of 2 µL of the sample in hexane were injected into the column with the carrier gas (hydrogen) flux at 1 mL/min and the split ratio was 1:10. The run was carried out in constant pressure mode. The oven temperature was held at 250 °C for 1 min, and increased to 260 °C over 20 min at the rate of 0.5 °C/min, and then increased to 325 °C over 13 min at the rate of 5 °C/min, and kept at 325 °C for 15 min. The injector and the detector temperatures were set at 325 °C. Chromatograms were recorded with LabSolution (Shimadzu, Columbia, MD, USA). Sterols were calculated by comparing the area of the samples and internal standards and expressed as mg/100 g of cheese.

#### 2.2.4. Phenol Extraction, Quantitation and Characterization

A liquid–liquid extraction was used to isolate the phenolic fraction from the cheeses and kiwifruit, following Suarez et al. [21] with some modifications. Briefly, 10 mL of methanol/water (80/20, *v/v*) were added to 5 g of sample and homogenized for 2 min with an ULTRATURRAX (IKA^®^-Werke GmbH & Co. KG, Staufen, Germany). After this, two phases were separated by centrifugation at 637× *g* for 10 min and the supernatant (hydroalcoholic phase) was transferred to a balloon. This step was repeated with 5 mL of methanol and the extracts were combined in the balloon. The hydroalcoholic extracts were then rotary evaporated to a syrupy consistency at 31 °C and dissolved in 5 mL of acetonitrile. Subsequently, the extract was washed with 10 mL of n-hexane and the rejected n-hexane was treated with 5 mL of acetonitrile. The acetonitrile solution was finally rotary evaporated to dryness. It was then re-dissolved in 5 mL of methanol and maintained at −20 °C before the chromatographic analysis.

Total phenol concentration extracts were determined spectrophotometrically by the Folin–Ciocalteu assay [22] using gallic acid as a standard. An aliquot of 1 mL of each extract was mixed with 5 mL of H_2_O and 1 mL of Folin–Ciocalteu phenol reagent 1N. The reaction had a duration of 7 min. A total of 10 mL of saturated Na_2_CO_3_ solution (7.5%) and 5 mL of H_2_O were then added and allowed to stand for 90 min before the absorbance of the reaction mixture was measured in triplicate at 750 nm. The total phenol content was expressed as mg of polyphenols per 100 g of cheese.

Individual polyphenol profiles by HPLC analysis were determined according to Kim et al. [23] with slight modifications. Briefly, 20 μL of each sample were analysed using a Prostar HPLC (Varian) with UV-DAD and a C18 reverse phase column (ChromSep HPLC Columns SS 250 mm × 4.6 mm including Holder with ChromSep guard column Omnispher 5 C18). The PDA acquisition wavelength was set in the range of 200–400 nm, with an analogue output channel at wavelength 280 nm width 10 nm. The gradient elution was performed by varying the proportion of solvent A (water–acetic acid, 97:3 v/v) to solvent B (methanol), with a flow rate of 1 mL min−1. The initial mobile phase composition was 100% solvent A for 1 min, followed by a linear increase in solvent B to 63% in 27 min. The mobile phase composition was then brought back to the initial conditions in 2 min for the next run. All the solutions prepared were filtered through 0.45 μm membranes.

#### 2.2.5. Volatile Organic Compounds Analysis

The volatile organic compounds were determined by solid phase microextraction–gas chromatography–mass spectrometry (SPME-GC/MS), according to Serra et al. [24]. Briefly, volatile organic compounds (VOCs) were extracted from 5 g of a finely-ground sample in a 20-mL glass vial closed with an aluminium cap equipped with a PTFE-septum. Samples were incubated for 15 min and then VOCs were collected using a divinylbenzene/carboxen/polydimethylsiloxane (DVB/Carboxen/PDMS) Stable Flex SPME fibre (50/30 μm; 2-cm long) (Supelco, Bellefonte, PA, USA). The SPME fibre was exposed to headspace for 30 min. The conditioning and exposure were carried out at 60 °C [25]. The fibre was inserted into the injector of a single quadrupole GC/MS (TRACE GC/MS, Thermo-Finnigan, Waltham, MA, USA) set at 250 °C, 3 min in splitless mode, keeping the fibre in the injector for 30 min to obtain complete fibre desorption.

The GC programme conditions were the same as those described by Serra et al. [24]. The GC was coupled with a Varian CP-WAX-52 capillary column (60 m × 0.32 mm; coating thickness 0.5 μm). The transfer-line and the ion source were both set at 250 °C. The filament emission current was 70 eV. A mass range from 35 to 270 *m/z* was scanned at a rate of 1.6 amu/s. The acquisition was carried out by electron impact, using the full scan (TIC) mode. Three replicates (*n* = 3) were run per sample. The VOCs were identified in three different ways: (i) comparison with the mass spectra of the Wiley library (version 2.0-11/2008); (ii) injection of authentic standards; and (iii) calculation of the linear retention index (LRI) and matching with reported indexes [26,27,28]. Data were expressed as the peak percentages of the total VOCs.

### 2.3. Statistical analysis

JMP software (SAS Institute Inc., Cary, NC, USA) was used for the statistical analysis. Data were analysed with the following mixed linear model:y_ij_ = µ + C_i_ + B_j_ (C_i_) + *e*_ij_(1)
where y_ij_ = the dependent variables (physico-chemical component, lactodimography data, fatty acids, sterols, polyphenols) relative to the i^th^ coagulant and to the j^th^ batch; μ = the mean; C_i_ = the fixed effect of the i^th^ coagulant (BM-C vs. BM-K or SM-C vs. SM-K); R_j_ (C_i_) = the random effect of the j^th^ batch (1 or 2) nested within C_i_; and *e*_ij_ = the random residual.

## 3. Results

### 3.1. Technological Parameters

The milk-clotting activity could only be assessed visually, as the r value could not be detected (higher than 30 min) either for BM-K or SM-K (Table 1).

The first appearance of solid material after adding the kiwifruit extract was observed after 20–22 min and 15–17 min in buffalo and sheep milk, respectively. These times were significantly higher than the milk with rennet calf (Table 1).

The kind of coagulant was a significant variation factor with respect to whey volume, and to initial cheese yield (Table 1). These differences became smaller after 24 h (rennet cheese weight decreased more than the kiwi extract) but the cheese yield was still significant.

### 3.2. Physico-Chemical Composition

The chemical compositions and colour characteristics of the cheeses are reported in Table 2. Dry matter was on average 40% similar to a typical fresh cheese. Buffalo cheeses showed a lower protein, fat and ash content than sheep cheeses.

The kiwi extract produced a cheese with a higher dry matter, both in buffalo and sheep milk (Table 2), except for proteins, which were not affected by the different holding whey capacity.

Iron, magnesium, potassium and calcium were not affected by cheesemaking. The kind of coagulant affected the sodium content of the cheese. Cheeses produced with the kiwifruit extract had the lowest amount of sodium (-34% and -14% in buffalo and sheep cheese, respectively).

The kiwi extract significantly affected the colour of the cheese (Table 2), reducing the lightness both in sheep and buffalo cheese. The use of kiwi also increased the b* value in buffalo cheese, but not in sheep cheese. Total colour differences (E) were less than 2.

### 3.3. Polyphenols

Although polyphenols are water-soluble, most were found in the curd but very few in the whey (Figure 1).

Cheeses obtained with the kiwi extract showed a significantly higher content of polyphenols, both in buffalo (+466%) and sheep cheeses (+278%) (Table 3).

The polyphenol content of SM-C was high, because the sheep milk used for our experiments was obtained from grazing animals. Our results are in line with Hilario et al. [29] in cheese produced with milk from pasture fed goats.

Cheeses obtained with the kiwifruit extract were higher both in flavonoids (quercetin, rutin and catechin) and in phenolic acids (caffeic acid, coumaric acid and cinnamic acid). They also contained coumaric and cinnamic acids, which were not present in the calf rennet cheese. The only polyphenol not affected by the cheesemaking was gallic acid.

### 3.4. Sterols

The use of kiwi extract promoted the accumulation of a small (but not negligible) quantity of phytosterols: 1.00 and 3.81 mg/100 g of total lipids in buffalo and sheep cheeses, respectively. On the other hand, the sterol profile of the rennet cheeses was characterized by cholesterol alone (Table 3). The phytosterols we found in cheeses were stigmasterol, campesterol and β-sitosterol, which represent the most abundant sterols in kiwifruit [30].

### 3.5. Volatile Organic Compounds

As expected, kiwifruit and cheeses were different in terms of the quantity and quality of the odorant compounds (Table 4). The total amount of VOCs from the kiwifruits was about seven times higher than from the cheeses. The most represented category of odorants in kiwi were aldehydes (about 85% of total VOCs), and carboxylic acids and ketones in the cheeses (65% of total VOCs).

More specifically, (E)-2-hexenal and hexanal were the most abundant odorants in the kiwi extracts used in this experiment. These volatile aldehydes are produced during fatty acid oxidation by the lipo-oxygenase enzyme [31]. Other substances affecting odour were the ethyl esters of butyric and caproic acids, two alcohols (2-hexen 1-ol and 1-hexanol) and one ketone, 1-penten-3-one (Table 4).

In terms of cheese, the most represented odorant was 2-butanone (a ketone), which was above 25% of the total VOCs in all the cheeses, followed by acetic acid, 2,3-butenedione (another ketone), caproic, caprylic and nonanoic acids (carboxylic acids). 2-butanone and 2,3-butenedione were not detected in the kiwi pulp, while acetic, caproic, caprylic and nonanoic acids were identified both in the kiwi pulp and in the buffalo and sheep cheeses produced, and both using calf rennet and kiwifruit extract. On the other hand, caproic, caprylic and nonanoic acids are typical components of both buffalo [32] and sheep milk [33].

Terpenes are typical components of kiwi pulp but not of cheese (except for β-phellandrene).

Of the odorants that most characterize kiwifruit, in both buffalo and sheep cheeses we found only (E)-2-hexenal (representing about 80% of VOCs in pulp kiwifruit), ethyl caproate and 2-hexen-1-ol. Interestingly, in the kiwi-cheeses we found 3-methyl eicosane and dibutyl formaldehyde, which were found only in low quantities in the kiwi extract. Finally, cheesemaking was a significant variation factor for the total esters only in buffalo cheese, 3-pentanone 2-hydroxy and β-phellandrene, which was the only terpene we detected in the cheeses.

## 4. Discussion

### 4.1. Tecnological Parameters

Table 1 shows that the technological parameters were affected by the coagulant. As demonstrated by the mL of whey released (Table 1) and by the amount of total solid (Table 2), the higher initial cheese yield from calf rennet was due to a higher whey holding capacity and, consequently, a lower and slower curd syneresis. This result is due to the different proteinase action between actinidine and chymosin. Chymosin is an aspartate proteinase that hydrolyses the Met105–Phe106 linkage of K-casein, cleaving out glycol-macropeptide. This leads to a decreasing polarity of the casein micelle and thus promotes the coagulation of casein. On the other hand, actinidin is a cysteine proteinase that cleaves bonds with basic amino acids in position P1, especially in β-casein, with a consequent reduction in milk clotting [5]. Thus, chymosin hydrolysed k-casein only, generating a more elastic and structured curd. Actinidin, instead, cleaves bonds in different sites, producing a curd with small peptones and with a more rapid syneresis.

### 4.2. Physic-Chemical Composition

The higher syneresis of both buffalo and sheep curd from kiwi extract was responsible for the higher dry matter of these cheeses compared to those obtained by calf rennet (Table 2). Thus, the higher content of lipids and ashes in BM-K and in SM-K are due to a “concentration” effect. However, the cheese protein content was not affected by the type of coagulant used. This could be due to the higher proteolysis of cheeses obtained with kiwi extract, which produces small peptide fragments (5b), which, in turn, are soluble in whey, thus offsetting the wider syneresis of these kinds of cheeses.

The level of carbohydrates was expected to be lower in cheeses from kiwifruit, as these lose whey to a higher extent. In reality the level was higher than expected, perhaps because kiwi extract provided some sugars.

The mineral composition was in line with Cichoscki et al. [34] (Table 2). In spite of the wider syneresis of cheeses from kiwi and their solubility in water, except for sodium, none of the coagulants affected the mineral content of the cheese. Again, these minerals came to some extent from the kiwifruit extract. The fact that calcium was affected by the cheesemaking procedure is difficult to explain, as it is present in milk in three different forms: ionic, soluble as calcium phosphate and colloidal in apatite bridges within casein micelle.

The kind of coagulant affected the sodium content of the cheeses. In fact, cheeses produced by kiwifruit extract showed the lowest sodium amount. Calf rennet is rich in sodium chloride and sodium benzoate, which are added as a preservative, thus partially explaining the different amount of sodium in the cheeses.

The different proteolytic activity of the two kinds of coagulant may explain the differences in colour of the cheese. In fact, both the lower lightness and the higher yellow index, might be related to the occurrence of proteolysis, which, in turn, is related to cheese browning [35]. Is worth noting that the colour differences between two coagulants were detectable only using instruments, as the total colour differences (∆E) were much lower than 3, which is the discrimination threshold for the human eye [19].

### 4.3. Polyphenols

The polyphenol content of BM-K and SM-K was very high, as 100 g of these kinds of cheeses have a similar quantity of polyphenols as edible fruits and vegetables such as oranges (217 mg/100 g) and broccoli (290 mg/100 g) [36].

Compared to calf rennet, the kiwifruit extract produced cheeses that were more than 4.5 and 2.7 times richer in polyphenols in buffalo and sheep cheese, respectively. This difference is due not to the total amount of polyphenols (which was almost the same in BM-K and SM-K), but to the kind of milk used for cheesemaking (the sheep milk comes from grazing animals). This is particularly true for gallic acid, which was not negatively affected by cheesemaking. Gallic acid is one of the most abundant polyphenolic substances in plants used for grazing [37] and is not present in the kiwi fruit extract [38].

Coumaric and cinnamic acid were not found in the calf rennet cheese. They can thus be used as good proxies to assess the benefits of using kiwifruit in cheese manufacturing and to conclude that using kiwifruit in cheese coagulation helps to improve the functional features of cheese. In fact, the positive effects on human health of polyphenols are well known as they are able to fight cancer, diabetes, aging, hypertension, asthma and cardiovascular diseases [39]. They also protect against the oxidation of LDL cholesterol and other lipids in the blood [40].

Although polyphenols are water-soluble, most were found in the curd but very few in the whey (Figure 1). It is well known that polyphenols bind caseins using hydrogen bonds [41], thus becoming insoluble in water [42]. The interaction between the polyphenols and proteins is affected by the pH, temperature, phenolic structure and amino acid profile [43,44,45,46,47], and represents a very interesting means to enrich cheese with polyphenols, and, consequently, to increase the nutritional and functional characteristics of cheese [42].

### 4.4. Sterols

The cheese produced using the kiwi extract contained not only cholesterol (the typical sterol of animal fat) but also some phytosterols, such as stigmasterol, campesterol andβ-sitosterol. These substances have several benefits for human health [48,49,50,51,52,53]. The level of phytosterols observed in BM-K and SM-K cheeses was insufficiently high to obtain a significant reduction in cholesterol absorption (2–3 g/die), or a corresponding reduction in the blood level of LDL-cholesterol (about 6%–15%) [53]. However, they are another positive feature of cheese obtained by kiwifruit coagulant, which helps to improve the overall nutritional and nutraceutical properties. Again, the presence of phytosterols in cheese may be a proxy to trace kiwi extract as a coagulant.

### 4.5. Volatile Organic Compounds

It was worth studying the effect of a kiwifruit extract on the volatile organic compounds (VOCs) of cheese, since these substances determine the taste and flavour of cheese, and thus influence the consumer’s choice. The key odorants of *Actinidia deliciosa* are (E)-2-hexenal, hexanal, ethyl butyrate and 1-penten-3-one, which give a herbal, sweet, marzipan odour [31]; in addition, ethyl butyrate, ethyl caproate, 2-hexen 1-ol and 1-hexanol and 1-penten-3 are contained in non-negligible quantities in kiwifruit (Table 4). They are fat soluble substances and thus we expected them to have been transferred to the cheese during cheesemaking, negatively affecting its organoleptic properties. However, we found that only (E)-2-hexenal (which represents about 80% of VOCs in kiwifruit extract), ethyl caproate and 2-hexen-1-ol, as well as two others volatile substances, 3-methyl eicosane and dibutyl formaldehyde, with a lower content in the kiwi extract, were transferred from kiwifruit into the buffalo and sheep cheese. These substances were found in the “kiwi cheeses” in quantities never higher than 2 µg/kg of cheese and in total accounted for less than 3.4 µg/kg, making up less than 2.5% of the total VOCs of both cheeses.

On the other hand, 2-butanone 2,3-butenedione made up over 30% of the total VOCs in all the cheeses and thus were not impacted by the cheesemaking. Given that the substances that most affected the aroma of the cheeses were the same, irrespectively of the cheesemaking procedure, and that the typical odorants of the kiwi aroma were transferred to the cheese to a very low extent, this would seem to indicate that cheesemaking with the kiwi extract led to a transfer of some volatile substances into the cheese. Further and specific organoleptic tests to assess whether this affects the aroma and taste of cheese should to be done. Finally, VOCs can be effective in tracing of the cheesemaking process.

## 5. Conclusions

Curd from kiwifruit showed a higher syneresis, giving a lower cheese yield (both initially and after 24 h) than calf-rennet.

The kiwifruit extract improved the nutraceutical properties of cheese by increasing the amount of polyphenols, which were 4.5 times (buffalo) and 3 times (sheep) higher than in cheese made with calf-rennet and phytosterols, which were only detected in cheese obtained with kiwifruit extract. These characteristics represent an important opportunity to produce cheese with a better nutraceutical quality. Finally, cheesemaking with the kiwi extract led to a transfer of some volatile substances into the cheese; specific tests to assess whether this affects the organoleptic properties of cheese are needed.

The results of this research highlight the possibility of using kiwifruit extract as an alternative to rennet in the coagulation of milk.

## Figures and Tables

**Figure 1 foods-09-00637-f001:**
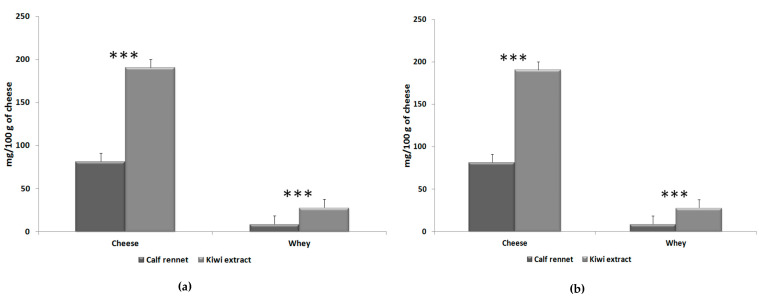
Total polyphenol content of cheese and whey produced with kiwifruit extract and calf rennet. (**a**) Buffalo milk; (**b**) sheep milk. *** *P* < 0.001

**Table 1 foods-09-00637-t001:** Technological parameters of the cheeses made with kiwi extract or calf rennet.

	Buffalo					Sheep			
	BM_C_	BM_K_	SEM	S		SM_C_	SM_K_	SEM	S
Yield (%)	51.25	33.18	2.82	**		24.90	21.27	1.10	*
Whey volume (mL)	84.67	120.67	7.12	**		130.83	143.83	1.77	***
Yield after 24 h (%)	38.95	27.85	1.71	**		23.15	20.52	0.87	*
Weight reduction (%)	23.78	15.18	1.83	**		6.73	3.52	0.87	*
pH whey 24 h	5.23	5.16	0.14	ns		4.62	4.92	0.02	ns
Clotting time (min)	13.50	21.00	0.23	***		11.50	16.50	0.22	***
r (min)	10.96	n.r.	3.7	ne		7.38	n.r.	6.03	ne
k20 (min)	1.95	n.r.	0.41	ne		1.33	n.r.	1.66	ne
a30 (mm)	42.47	n.r.	6.20	ne		41.67	n.r.	2.86	ne

BMC, buffalo–calf rennet cheese; BMK, buffalo–kiwifruit cheese; SMC, sheep–calf rennet cheese; SMK, sheep–kiwifruit cheese; SEM, standard error medium; S, significance. *, 0.01 < *P*< 0.05; **, 0.01 < *P* < 0.001; ***, *P* < 0.001; ns: not significant; n.r., not reactive; ne: not estimable.

**Table 2 foods-09-00637-t002:** Physico-chemical composition of the cheeses made with kiwi extract or calf rennet.

	Buffalo					Sheep			
	BM_C_	BM_K_	SEM	S		SM_C_	SM_K_	SEM	S
Total solid (g/100 g)	35.89	44.44	1.83	^+++^		43.68	50.69	0.46	^+++^
Proteins (g/100 g)	10.76	11.74	0.26	^+^		15.83	15.74	0.23	ns
Lipids (g/100 g)	20.60	27.40	1.24	^++^		21.72	27.55	0.36	^+++^
Ashes (g/100 g)	0.99	1.27	0.03	^+++^		1.33	1.69	0.05	^+++^
Carbohydrates	4.05	4.53	0.75	ns		5.16	5.52	0.47	ns
Fe (μg/g)	0.71	0.81	0.09	ns		1.57	1.93	0.09	ns
Mg (μg/g)	266.01	254.73	14.05	ns		268.27	271.63	4.58	ns
K (μg/g)	1721.61	2136.11	132.05	ns		1727.81	1696.26	56.39	ns
Ca (μg/g)	7985.18	7938.71	259.73	ns		8131.28	8689.25	285.27	ns
Na (μg/g)	522.83	346.34	42.47	^+^		1173.25	1004.39	23.37	^+^
Colour									
L*	93.73	92.15	0.37	^+++^		93.12	92.09	0.24	^+^
a*	−1.75	−1.83	0.04	ns		−1.88	−1.73	0.05	ns
b*	8.68	10.62	0.45	^+^		12.17	12.33	0.17	ns
E*	1.84		0.68				1.05		0.36

BMC, buffalo–calf rennet cheese; BMK, buffalo–kiwifruit cheese; SMC, sheep–calf rennet cheese; SMK, sheep–kiwifruit cheese; ΔE between cheese from calf rennet—kiwifruit, ΔE* = ((ΔL*)2 + (Δa*)2 + (Δb*)2)1/2. Values are expressed as the mean ± standard deviation; SEM, standard error medium; S, significance. ^+^, 0.01< *P*< 0.05; ^++^, 0.01 < *P* < 0.001; ^+++^, *P* < 0.001; ns: not significant.

**Table 3 foods-09-00637-t003:** Polyphenol and sterol content of cheeses made with kiwi extract or calf rennet (mg/100 g of cheese).

	Buffalo					Sheep			
	BM_C_	BM_K_	SEM	S		SM_C_	SM_K_	SEM	S
Total polyphenols	32.13	149.94	7.65	***		68.02	189.28	13.40	***
Gallic acid	15.39	15.76	1.99	ns		34.24	15.18	4.97	*
Caffeic acid	4.16	35.97	2.72	*		6.32	11.52	1.50	*
Coumaric acid	-	21.60	1.93	ne		-	33.35	6.32	ne
Cinnamic acid	-	10.78	1.31	ne		-	27.33	6.08	ne
Quercetin	2.93	9.19	0.91	*		10.83	10.03	2.25	ns
Catechin	5.49	24.03	4.44	*		5.17	43.49	3.28	*
Rutin	4.15	43.85	2.91	***		-	58.61	9.85	ne
Sterols									
Cholesterol	5.71	5.51	0.14	ns		5.96	5.76	0.18	ns
Stigmasterol	-	0.17	0.00	ne		-	0.83	0.04	ne
Campesterol	-	0.20	0.00	ne		-	0.69	0.02	ne
β-sitosterol	-	0.63	0.02	ne		-	2.29	0.15	ne

BMC, buffalo–calf rennet cheese; BMK, buffalo–kiwifruit cheese; SMC, sheep–calf rennet cheese; SMK, sheep–kiwifruit cheese; SEM, standard error medium; S, significance. *: 0.01 < *P* < 0.05; **: 0.01 < *P* < 0.001; ***: *P* < 0.001; ns: not significant; ne: not estimable.

**Table 4 foods-09-00637-t004:** Volatile organic compounds of cheeses made with kiwifruit extract or calf rennet (µg/kg of raw matter) (Part 1).

	Kiwi			Buffalo				Sheep			
			BM_C_	BM_K_	SEM	S		SM_C_	SM_K_	SEM	S
**Acids**											
Acetic acid	4.00		9.06	9.06	2.58	ns		2.58	3.03	0.43	ns
Butyric acid	1.56		2.26	1.58	0.48	ns		2.72	3.27	0.74	ns
Caproic acid	3.22		6.28	3.63	1.34	ns		7.65	9.65	2.38	ns
heptanoic acid	1.31		0.95	1.11	0.16	ns		0.79	0.57	0.07	*
caprylic acid	2.31		6.17	7.23	2.88	ns		10.53	13.95	2.83	ns
n-nonanoic acid	3.55		4.97	3.34	1.04	ns		3.83	4.83	0.83	ns
n-decanoic acid	1.69		1.78	2.61	0.76	ns		5.36	8.21	1.60	ns
*Total*	*17.64*		*31.46*	*28.56*	*6.77*	*ns*		*33.46*	*43.50*	8.21	ns
**Alcohols**											
2-methyl-3-pentanol	-		1.00	2.03	0.28	*		2.15	0.58	0.14	***
2-methyl-1-undecanol	-		1.39	1.98	0.53	ns		1.83	0.90	0.13	***
2-buthyl-1-octanol	-		0.77	0.81	0.19	ns		1.65	0.88	0.35	ns
1-pentanol	2.92		0.51	0.49	0.06	ns		0.87	1.29	0.16	*
1-hexanol	11.12		1.31	0.91	0.16	ns		0.67	0.91	-	ns
3,4-hexane diol	-		3.31	6.11	1.26	ns		3.67	9.52	0.98	**
3-hexen-1-ol	1.67		-	-	-	-		-	-	-	-
2-hexen 1-ol	12.96		-	0.34	0.05	ne		-	0.46	0.07	ne
1-octen-3-ol	0.99		0.30	0.49	0.03	**		0.11	0.15	0.01	*
2,4,7,9-tetramethyl-5-decyne-4,7-diol	1.49		2.84	0.82	0.23	***		1.61	2.73	0.77	ns
*Total*	*31.11*		*11.42*	*13.97*	*1.64*	*ns*		*12.56*	*17.41*	1.79	ns
**Aldehydes**											
Acetaldehyde	1.65		-	-	-	-		0.16	0.21	0.09	ns
Hexanal	24.48		3.82	3.22	1.01	ns		3.21	2.67	0.70	ns
(*E*)-2-hexenal	710.37		-	1.06	0.09	ne		0.09	1.82	0.26	***
2-heptenal	0.52		-	-	-	-		-	-	-	-
Nonanal	6.66		0.49	0.55	0.13	ns		0.57	0.63	0.07	ns
2-octenal	1.79		-	-	-	-		-	-	-	-
Decanal	0.89		-	-	-	-		-	-	-	-
*trans*-2-decenal	2.11		-	-	-	ns		-	-	-	-
Undec-2-enal	1.50		-	-	-			-	-	-	-
Dibutyl formaldehyde	7.12		-	0.48	-	ne		-	0.21	0.02	ne
*Total*	*757.09*		*4.31*	*5.31*	*1.17*	*ns*		*4.03*	*5.54*	*0.88*	*ns*
**Alkanes and Alkenes**											
2,2 dimethyl decane	-		1.07	1.21	0.37	ns		1.53	1.01	0.34	ns
2,5,6-trimethyldecane	-		2.90	3.79	0.61	ns		4.86	2.79	0.66	*
2,5-dimethylnonane	-		1.52	-	0.01	ne		1.77	-	0.16	ne
2,6,11-trimethyldodecane	-		1.45	-	0.04	ne		0.85	-	0.12	ne
2,5-dimethylundecane	-		1.40	1.14	0.37	ns		1.29	1.07	0.33	ns
2,3 dimethyl nonane	-		1.13	1.32	0.28	ns		1.53	1.30	0.26	ns
3 methyl decane	1.16		1.46	0.67	0.18	*		1.27	0.98	0.24	ns
3-methyl eicosane	5.19		-	0.80	0.01	ne		-	0.75	0.03	ne
5-methyl-undecane	1.65		0.60	0.54	0.08	ns		0.38	0.82	0.07	**
4,5 dipropyloctane	-		1.19	2.11	0.31	*		1.71	1.06	0.30	ns
5-ethyl decane	-		2.30	1.99	0.47	ns		2.27	2.24	0.39	ns
3,5-dimethyl undecane	1.55		0.46	0.47	0.10	ns		0.27	0.34	0.04	ns
2,4-dimethyl-1-heptene	-		1.11	1.64	0.35	ns		3.11	1.68	0.80	ns
5-methyl-1-undecene	-		1.69	-	0.53	ne		1.73	-	0.11	ne
*Total*	*9.55*		*15.76*	*13.20*	*1.47*	*ns*		*19.74*	*11.66*	*2.26*	***
**Aromatic hydrocarbons**											
p-xylene	-		1.28	2.16	0.97	ns		2.34	1.40	0.57	ns
o-xylene	-		0.29	1.79	0.61	ns		1.65	1.09	0.47	ns
*Total*	*-*		*1.57*	*3.95*	*1.54*	*ns*		*3.99*	*2.48*	*1.03*	*ns*
**Esters**											
Ethyl acetate	7.86		0.50	0.54	0.06	ns		0.88	0.55	0.14	ns
Ethyl butyrate	11.98		-	-	-	-		-	-	-	-
Ethyl caproate	11.43		-	0.42	0.01	ne		-	0.15	-	-
Ethyl caprilate	1.39		-	-	-	-		-	-	-	-
n-heptyl formate	1.84		0.42	0.37	0.06	ns		-	-	-	-
*Total*	*34.50*		*0.92*	*1.33*	*0.07*	****		*0.88*	*0.70*	*0.13*	*ns*
**Ketones**											
2,3-butenedione	-		7.66	6.53	1.11	ns		4.94	5.17	0.72	ns
1-penten-3-one	2.05		-	-	-	-		-	-	-	-
2,3 pentanedione	-		4.70	5.20	0.77	ns		2.82	2.23	0.41	ns
2-butanone	-		30.77	38.30	4.69	ns		38.93	40.90	5.36	ns
3-pentanone 2-hydroxy	-		2.14	3.90	0.64	*		2.32	5.70	0.54	**
2-nonanone	-		0.66	0.43	0.07	ns		0.66	0.77	0.12	ns
*Total*	*2.05*		*46.42*	*52.66*	*6.62*	*ns*		*49.54*	*55.72*	*6.29*	*ns*
**Terpenes**											
β-phellandrene	2.95		0.12	0.25	0.01	***		0.45	0.66	0.04	**
m-cymene	1.79		-	-	-	-		-	-	-	-
p-mentha-1,4(8)-diene	2.26		-	-	-	-		-	-	-	-
Pinocanphone	2.61		-	-	-	-		-	-	-	-
3-pinanone	6.21		-	-	-	-		-	-	-	-
2-pinen-4-one	4.64		-	-	-	-		-	-	-	-
*Total*	*20.46*		*0.12*	*0.25*	*0.01*	*****		*0.45*	*0.66*	*0.04*	****
**Others**											
2-ethyl-hexyl tert-butyl ether	-		1.44	1.92	0.28	ns		1.28	0.96	0.21	ns
Dimethyl disulfide	-		0.57	0.95	0.24	ns		-	-	-	-
2 ethyl hexyl chloroformate	-		2.95	6.15	2.80	ns		2.81	3.61	0.83	ns
a-ethyl-furan	8.12		-	-	-	-		0.42	0.33	0.09	ns
m-d-tert-butyl-benzene	-		0.26	0.45	0.09	ns		0.71	0.60	0.15	ns
Ethylhexanol	4.83		-	-	-	-		-	-	-	-
1-cyclopropyl pentane	2.33		0.23	0.22	0.04	ns		0.39	0.33	0.05	ns
1-hexyl-2-methylcyclopropane	1.78		0.42	0.44	0.10	ns		0.40	0.23	0.03	ns
Ethyl benzene carboxylate	3.88		-	-	-			-	-	-	-
*Total*	*20.94*		*5.85*	*10.11*	*2.48*	*ns*		*6.01*	*6.06*	*1.13*	*ns*
**Total VOCs**	893.34		117.85	129.34	15.61	ns		130.88	143.55	11.91	ns

BMC, buffalo–calf rennet cheese; BMK, buffalo–kiwifruit cheese; SMC, sheep–calf rennet cheese; SMK, sheep–kiwifruit cheese; SEM, standard error medium; S, significance; ns, not significant; ne, not estimable. *: 0.01 < *P* < 0.05; **: 0.01 < *P* < 0.001; ***: *P* < 0.001;

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
