# Peer review of "Nutraceutical and Technological Properties of Buffalo and Sheep Cheese Produced by the Addition of Kiwi Juice as a Coagulant"

_foods, 2020, doi:10.3390/foods9050637_

Round 1

Reviewer 1 Report

The objective of this work was to study the effect of kiwifruit enzymatic extract as milk coagulant on the technological and physico-chemical properpies of ewe and buffalo fresh cheeses compared with those made from animal rennet.

In general, this is a confirmatory study on the research area (vegetable clotting enzymes versus animal rennet properties in cheesemaking), and the main criticisms concern on (1) the number of independent repetitions made of the experiment, on (2) the coagulanting activity of kiwifruit extract used which could determine the technological properties, and on (3) the microbiological quality of cheeses.

In my opinion, the paper needs further work.

Introduction

Line 27. There exits other types of milk coagulation with no addition of clotting enzymes. Sentence could be: Enzymatic milk coagulation is …

Introduction could be improved by adding relevant information obtained by other authors working on kiwifruit extracts.

Material and methods

Lines 58-62. Authors must describe how many independent productions were made, more than to describe how many cheeses.

Line 70. Cheeses were made from 200 mL of milk? Are they model cheeses?

Lines 72-74. Why an aliquot of lyophilized kiwifruit extract containing 20 g of protein was added? Suspended in 2 mL of water? Was standardized the IMCU per litre in the calf rennet and kiwifruit extract to cheesemaking? If coagulanting activities of enzyme extracts were not standardized it is normal the differences noted by the authors on the coagulation times. Normally, the coagulating activities are standardized and then the curd characteristics, syneresis, cheese yield and components (protein and fat) lost in cheese whey are studied.

Lines 75-79. Please, explain the different steps of cheesemaking: cutting, draining, moulding, pressing, salting, …

Line 87. How many independent determinations were made? Only calf rennet was evaluated by Formagraph? And the kiwifruit extract? Please add conditions used with kiwifruit extract. I do not understand why not coagulation conditions used during cheesemaking were used in the Formagraph assay.

Line 116. What is the objective for the total fatty acid analyses?

Lines 163-172. Analysis reference?

Line 192. Statistical analysis must explain the experiment repetitions made.

Table 1. Calculate the cheese yield and cheese yield corrected by the humidity at 24 h more than the initial weight. Add the units of Clotting time. If only Formagraph results were obtained with calf rennet, these results are not transcendent in the study. Please, calculate the colour differences between cheeses by using ΔE parameter. Using this parameter it is possible to assess the real colour differences between cheeses.

Line 221. Change then for than.

Lines 219-221. Rewrite sentence.

Table 2. Change Dry matter by Total solids.

Table 3. Add Sheep on the type of sheep cheeses.

Line 261. Why fatty acid composition of milk triglycerides would be changed by the coagulant?

Line 303. Rewrite sentence. Calf rennet had a higher weight?

Line 306. And this is positive or negative for this type of cheeses?

Lines 315-317. Colour is not a technological parameter. Please change to physico-chemical composition.

Lines 320-322. Lipids and ashes could be calculated in total solid basis. If cheese protein is expressed in total solid basis possibly differences would  be appreciated. Anyway, rewrite sentence… “However, cheese protein content was not affected by the type on coagulant used. This could be due…”

Line 326. Lower?

Line 330-332. The mineral come from kiwifruit, then are not soluble in water and remain in cheese? Cheese calcium was not affected by the type of coagulant used.

Lines 334-336. This is the same sentence that in results. Please, not repeat. Really, do you think that the small amount of rennet solution used increased the Na level in cheese? On the other hand, Na is soluble in water and a great quantity is lost in cheese whey.

Line 338. How the kiwifruit coagulant will be preserved as commercial coagulant?

Line 351. Change are by were.

Line 391. Sensory properties of cheese were assessed?

Line 394. Please, add the most important  findings of the study.

Author Response

The objective of this work was to study the effect of kiwifruit enzymatic extract as milk coagulant on the technological and physico-chemical properpies of ewe and buffalo fresh cheeses compared with those made from animal rennet.

In general, this is a confirmatory study on the research area (vegetable clotting enzymes versus animal rennet properties in cheesemaking), and the main criticisms concern on (1) the number of independent repetitions made of the experiment, on (2) the coagulanting activity of kiwifruit extract used which could determine the technological properties, and on (3) the microbiological quality of cheeses.

In my opinion, the paper needs further work.

AU: Thanks a lot for dedicating your time to review of our work. We greatly appreciated your suggestions and we attempted to reply to your remarks. You’ll can find the text modifications in red. We hope the quality of paper is now improved.

Introduction

Line 27. There exits other types of milk coagulation with no addition of clotting enzymes. Sentence could be: Enzymatic milk coagulation is …

AU: sentence changed

Introduction could be improved by adding relevant information obtained by other authors working on kiwifruit extracts.

AU: we provided this information also according to request of review 2

Material and methods

Lines 58-62. Authors must describe how many independent productions were made, more than to describe how many cheeses.

AU: You are right; sorry for the incomplete information: the experiment was replicate two times. we changed sentence and (see below) modified the formula of statistical model.  

Line 70. Cheeses were made from 200 mL of milk? Are they model cheeses?

AU: yes; we done cheesemaking in order to obtain “miniature cheese” (according to Mozorra-Manzano et al. 2013). We included this paper in the reference list.

Lines 72-74. Why an aliquot of lyophilized kiwifruit extract containing 20 g of protein was added? Suspended in 2 mL of water? Was standardized the IMCU per litre in the calf rennet and kiwifruit extract to cheesemaking? If coagulanting activities of enzyme extracts were not standardized it is normal the differences noted by the authors on the coagulation times. Normally, the coagulating activities are standardized and then the curd characteristics, syneresis, cheese yield and components (protein and fat) lost in cheese whey are studied.

AU: Sorry, 20 g of protein is a typo; obviously we arrange the experiment standardizing the IMCU between the calf rennet and kiwi fruit. We rewritten the sentence; I hope this is now clearer what we done.  

Lines 75-79. Please, explain the different steps of cheesemaking: cutting, draining, moulding, pressing, salting, …

AU: We explained the cheesemaking. We hope the procedure we used is now clear enough.

Line 87. How many independent determinations were made? Only calf rennet was evaluated by Formagraph? And the kiwifruit extract? Please add conditions used with kiwifruit extract. I do not understand why not coagulation conditions used during cheesemaking were used in the Formagraph assay.

AU: we done the formagraph analysis for kiwifruit as well; we included the specification (apologies for the no complete information we had given).  We done milk coagulation with calf rennet according to formagraph instructions, then we have calculated the quantity of kiwi coagulant to obtain the same strength as calf rennet.

Line 116. What is the objective for the total fatty acid analyses?

AU: we done the fatty acid analysis to provide a complete characterization of cheeses; nevertheless, as we said at lines 265-267 we expected no significant results. Anyway, I can understand your remark, so we deleted the paragraph of fatty acid analysis.

Lines 163-172. Analysis reference?

AU: we include the reference.

Line 192. Statistical analysis must explain the experiment repetitions made.

AU: we included the batch in the statistical model

Table 1. Calculate the cheese yield and cheese yield corrected by the humidity at 24 h more than the initial weight. Add the units of Clotting time. If only Formagraph results were obtained with calf rennet, these results are not transcendent in the study. Please, calculate the colour differences between cheeses by using ΔE parameter. Using this parameter it is possible to assess the real colour differences between cheeses.

AU: We calculated cheese yield and cheese yield after 24 h and we included the ΔE for the evaluation of colour parameters (we included them in material and method as well). We included some sentences (in red) about these results in the paragraph 3.1. As we said above, we performed the formagraph analysis for kiwifruit coagulant as well. The table shows that for this kind of coaugulant the milk coagulation happened after 30 minutes, thus wasn’t detected by the instrument.

Line 221. Change then for than.

AU: changed

Lines 219-221. Rewrite sentence.

AU: we changed sentence and we made it different from that of the discussion.

Table 2. Change Dry matter by Total solids.

AU: changed

Table 3. Add Sheep on the type of sheep cheeses.

AU: added

Line 261. Why fatty acid composition of milk triglycerides would be changed by the coagulant?

AU: I agree with you; see aforementioned reply (we decided to delete this paragraph and S1 table).

Line 303. Rewrite sentence. Calf rennet had a higher weight?

AU: we rewrote sentence

Line 306. And this is positive or negative for this type of cheeses?

AU: we changed the sentence of the lines 304-308; we hope we had replied both to your remarks (line 303 and 306)

 Lines 315-317. Colour is not a technological parameter. Please change to physico-chemical composition.

AU: we moved the color parameters from table 1 to table 2 and colour comment from paragraph 3.1 to 3.2 (results) and from 4.1 to 4.2 (discussion)

Lines 320-322. Lipids and ashes could be calculated in total solid basis. If cheese protein is expressed in total solid basis possibly differences would be appreciated. Anyway, rewrite sentence… “However, cheese protein content was not affected by the type on coagulant used. This could be due…”

AU: Yes, you are right, but we chose to express data in absolute terms because we wanted to highlight the nutritional characteristics of cheeses. In this case we think this manner of data expression make reading easier. Anyway, we changed sentence as you requested.

Line 326. Lower?

AU: we changed sentence. We hope this it's now clearer.

Line 330-332. The mineral come from kiwifruit, then are not soluble in water and remain in cheese? Cheese calcium was not affected by the type of coagulant used.

AU: we think that the kiwifruit provided a lot of mineral; thus, there were not differences despite the higher way loss of curd from kiwifruit. For the calcium the hypothesis is too hard to be done, due to the different form a calcium in the milk.   

Lines 334-336. This is the same sentence that in results. Please, not repeat. Really, do you think that the small amount of rennet solution used increased the Na level in cheese? On the other hand, Na is soluble in water and a great quantity is lost in cheese whey.

AU: we changed sentence; I think that this could be an explanation as sodium was the only mineral affected by cheesemaking. Anyway, I can understand your remark, for this I deleted the last part of sentence, giving less importance to this conclusion.

Line 338. How the kiwifruit coagulant will be preserved as commercial coagulant?

AU: we think that the freeze dry of kiwifruit could be an effective preserving method

Line 351. Change are by were.

AU: done

Line 391. Sensory properties of cheese were assessed?

AU: sentence changed

Line 394. Please, add the most important  findings of the study.

AU: we changed the conclusion by adding the most important findings. 

Reviewer 2 Report

Generally, the topic of the manuscript is interesting.

But I do have two major concerns:

1) The experimental design:

Why did the authors choose these particular concentrations of the different coagulants? l.72-74

In the abstract it is stated that 'clotting properties have been widely investigated'. This information is missing in the manuscript (see also my comment ot the Introduction).

This resulted in major differences right from the start of the cheesemaking process (see Table 1) making comparisons difficult.

l.203-204: In my opinion it would have been very easy to determine the appropriate level of kiwi extract to improve comparability. If that is not the case, why then choosing chymosin as 'reference' and not another already utilized plant clotting enzyme?

In the Introduction part information on actinidin is very imprecise (l.47-51). More information from these cited papers might maybe help ('suitable' clotting agent; purified,...)

2) There is absolutely no information on the sensory and texture properties of the 'cheeses'. What kind of cheese did you intend to produce?  Desrciption is quite scarce and to be honest strange: l.75-79. Additionally, in Table 1 there are no parameters available from the Formagraph for 'cheeses' with kiwi extract. So kind of product did you get?

What are the 'organoleptic' tests you refer to?

Some other points:

Chemical analysis s quite excessive. What is the merit from Table 4? Wouldn't it be of more interest to combine the Results and Discussion parts and focus on the major results?

Author Response

Generally, the topic of the manuscript is interesting. 

AU: thanks for appreciating the topic of our paper. We attempted to reply to all your questions and remarks. We highlighted the new text in red.

But I do have two major concerns:

1) The experimental design:

Why did the authors choose these particular concentrations of the different coagulants? l.72-74

AU: we changed the sentence to accept the remarks of referee 1. In the sentence there were so error (apologies for this). We hope the explanation in now clearer. Anyway: we attempted to standardize the strength of two so different kind of coagulants, by starting to the vegetal one. We did some preliminary tests to assets the coagulant properties of the kiwi; then we have calculated the same strength for the calf rennet (by specific dilution)

In the abstract it is stated that 'clotting properties have been widely investigated'. This information is missing in the manuscript (see also my comment ot the Introduction).

AU: we provided more information about the actinidin

This resulted in major differences right from the start of the cheesemaking process (see Table 1) making comparisons difficult.

AU: As we said above, we used coagulants with a comparable strength, thus we think that the differences are due to different proteolytic activity of two coagulants. Our main intension was to stress the nutritional and nutraceutical characteristics of cheese

l.203-204: In my opinion it would have been very easy to determine the appropriate level of kiwi extract to improve comparability. If that is not the case, why then choosing chymosin as 'reference' and not another already utilized plant clotting enzyme?

AU: For the formagraph analysis we started from the calf- rennet, and we done it accordind to formagraf instructions. Then, we calculated the amount of kiwifruit to have the same strength. In our opinion, the lack in milk coagulation of kiwifruit with formagraph could be a demonstration of the worst coagulation aptitude of kiwifruit than calf rennet 

In the Introduction part information on actinidin is very imprecise (l.47-51). More information from these cited papers might maybe help ('suitable' clotting agent; purified,...)

AU: as we said above, we provided more information about the actinidin properties; we took this information both from the already cited papers and new references.

2) There is absolutely no information on the sensory and texture properties of the 'cheeses'. What kind of cheese did you intend to produce?  Desrciption is quite scarce and to be honest strange: l.75-79. Additionally, in Table 1 there are no parameters available from the Formagraph for 'cheeses' with kiwi extract. So kind of product did you get?

AU: sorry for giving incomplete information about the cheesemaking. honestly, the main objective of our work was to investigate the possibility to use kiwifruit as milk coagulant, focusing on the nutritional and nutraceutical characteristics of the cheese obtained.  Anyway, the experiment was done on a soft cheese; we provided a new description of the procedure we used and (I hope) it's clear enough now.

What are the 'organoleptic' tests you refer to?

AU: You are right. We didn’t perform organoleptic tests. We meant the volatile organic substances affecting the cheese aroma. We corrected the sentence

Some other points:

Chemical analysis s quite excessive. What is the merit from Table 4? Wouldn't it be of more interest to combine the Results and Discussion parts and focus on the major results? 

AU: Table 4 lists the VOCs and, in effect, includes a lot of data. On the other hand, we think that topic is very complex and that this table provide some new information about the transfer of substance affecting aroma from vegetable coagulant to milk and cheese. at the best of our knowledge, there are no paper showing this data. With respect to your second remark, we followed the template of the Journal.  

Round 2

Reviewer 1 Report

Authors have answered questions made by the reviewer and they have revised and corrected the manuscript.

Author Response

Thank you so much for your review and for accepting our answers.

Sincerely

Reviewer 2 Report

Please find my comments within the manuscript

Author Response

Thanks a lot for your revision. We attempted to reply to your requests. We included the number of the line of the modifications we made. In the text you’ll find other modification than the ones you asked us, because we sent the paper to a professional service for the language check.

Line 79 Is this concentration different to the formagraph experiments?

AU: the concentration is the same to the formagraph; we included this in the sentence at line 106

LINE 80. 80 mg for what? 1 mL?

AU we put 80mg in 200mL of milk (see line 76); anyway, we changed sentence (Line 81)

LINE 82. you mean manually?

AU: yes; sorry we changed sentence (Line 85)

LINE 117: This is not correct. Please have a closer look at the CIELAB colour space! e.g. a* from green (-) to red (+). Your a* values are negativ. Please be more correct.

AU: You are right; sorry for the error. sentence changed (line 121)

LINE 174: With respect, but the revision seems to be done in a hurry. Please check the entire manuscript for errors! There are quite a lot.

AU: we changed the name of the first author (Kim and not Kit) and we corrected the format of reference in the text according to the journal rules; we cheeked the whole paper and we made modification; sorry for the mistakes.

TABLE 4

  1. What is meant with ne?

I find it still difficult to extract valuable results from this large table. There are so many not significant differences which could be summarized in another way.

AU: “ne” means not estimable (we found this compound in cheeses made with kiwifruit only); we included the “ne” and “ns” in the caption of the table. We agree with you, Table 4 lists a lot of information, but this is the only way I know to show these data. In my opinion the few significant values mean that kiwi use affected VOCs profile of cheeses slightly.  

Line 354 This is not convincing. The amount of coagulant that is used for cheese-making is too low for having an effect on sodium content.

AU: you are right. I calculated carefully the amount of sodium coming from calf rennet, and this provide only a partial explanation for the differences in sodium content of cheeses. Thus, we changed the sentence (line 359). We changed the sentence about sodium in the conclusion paragraph as well. 

line 413: This is only a speculation. You can't state that.

AU: I understood your criticism, but we made a hypothesis considering the instrumental data. In the other hand I think that we have been “very cautious” and we said that this hypothesis has to be confirmed by specific tests (Lines 414-415). Any way we changed the sentence to make it more “hypothetic” (Lines 420-421). We change the sentence in the conclusion paragraph as well.

line 419 There are so many errors in this paragraph. Please out of respect for the journal, take your time to improve it.

AU: please accept my apologies; It wasn't my intention to disrespect Journal. I checked the paragraph and I attempted to improve it. Moreover, I deleted the sentence about sodium and aroma of cheese.